# Identification of Photocatalytic Alkaloids from Coptidis Rhizome by an Offline HPLC/CC/SCD Approach

**DOI:** 10.3390/molecules27196179

**Published:** 2022-09-21

**Authors:** Shu-Qin Qin, Jun Ma, Qi-Qi Wang, Wei Xu, Wen-Cai Ye, Ren-Wang Jiang

**Affiliations:** 1Guangdong Province Key Laboratory of Pharmacodynamic Constituents of TCM and New Drugs Research, College of Pharmacy, Jinan University, Guangzhou 510632, China; 2Research Institute of Guangdong HAID Group Co., Ltd., Guangzhou 511400, China

**Keywords:** co-crystallization, photocatalysis, alkaloids, HPLC/CC/SCD, nanoscale

## Abstract

Natural products continue to be a valuable source of active metabolites; however, researchers of natural products are mostly focused on the biological effects, and their chemical utility has been less explored. Furthermore, low throughput is a bottleneck for classical natural product research. In this work, a new offline HPLC/CC/SCD (high performance liquid chromatography followed by co-crystallization and single crystal diffraction) workflow was developed that greatly expedites the discovery of active compounds from crude natural product extracts. The photoactive total alkaloids of the herbal medicine Coptidis rhizome were firstly separated by HPLC, and the individual peaks were collected. A suitable coformer was screened by adding it to the individual peak solution and observing the precipitation, which was then redissolved and used for co-crystallization. Seven new co-crystals were obtained, and all the single crystals were subjected to X-ray diffraction analysis. The molecular structures of seven alkaloids from milligrams of crude extract were resolved within three days. NDS greatly decreases the required crystallization amounts of alkaloids to the nanoscale and enables rapid stoichiometric inclusion of all the major alkaloids with full occupancy, typically without disorder, affording well-refined structures. It is noteworthy that anomalous scattering by the coformer sulfur atoms enables reliable assignment of absolute configuration of stereogenic centers. Moreover, the identified alkaloids were firstly found to be photocatalysts for the green synthesis of benzimidazoles. This study demonstrates a new and green phytochemical workflow that can greatly accelerate natural product discovery from complex samples.

## 1. Introduction

Natural products are produced naturally by any organism including primary and secondary metabolites. In a broad sense, they include any substance produced by life. Worldwide revitalization has been taking place in the interest for natural product research [1]. In daily life, bombardment with advertisements for natural and organic products is very frequent. However, current natural product research still faces challenges, e.g., traditional research strategies and low efficient protocols. On one hand, researchers of natural products are mostly focused on the biological effects. Owing to the chemical diversity, target affinity, and specificity toward biological macromolecules, natural products derived from herbal medicines have been extensively applied in biological assays and have demonstrated enormous potential as modulators of biomolecular functions. The resulting lead compounds often play a significant role in drug discovery [2]. On the other hand, classical natural product research has low efficiency. Starting with a bulk quantity of raw material, through extraction and isolation, often results in micrograms of natural products [3]. Then, extensive spectroscopic analyses were used to elucidate the structures. The whole process consumes a large quantity of raw material and organic solvent and need months or years of time [4]. The high cost and low efficiency constitute major bottlenecks in the field of natural product chemistry, and development of new protocol is necessary. We reported herein the identification of photocatalytic alkaloids from Coptidis rhizome (CR) by an offline HPLC/CC/SCD (high performance liquid chromatography followed by co-crystallization and single crystal diffraction) approach and its applications as green photocatalysts for the synthesis of benzimidazoles.

The procedure for the offline HPLC/CC/SCD workflow is described in Figure 1. The first step was to produce a chromatogram from a crude natural sample by preparative HPLC, a separation technique that can selectively produce single peaks of natural compounds from crude extracts on a nanogram scale in a short time. In the second step, the individual peaks in the HPLC profile were collected and concentrated in vacuo. In the third step, the residue for the individual peaks were added a co-crystallization coformer 1,5-naphthalenedisulfonic acid (NDS) and subjected to crystal growth. Finally, the resulting co-crystals were analyzed by X-ray crystallographic analysis. After collection of diffraction data, the structures of the alkaloid guests were elucidated by X-ray crystallographic analysis.

## 2. Results

### 2.1. Photocatalysis of Benzimidazole Catalysed by the Raw Material and Total Alkaloids under Blue LED Irradiation

Benzimidazole and its derivatives are widely employed as privileged lead structures in drug discovery because of the unique structural features and a wide range of biological activities, e.g., anthelmintic [5], anti-tuberculosis [6], anti-hypertension [7], and antiproliferation [8]. *O*-phenylenediamine and benzaldehydes were used as substrates and dried CR powder was used as photocatalysts for the synthesis of benzimidazole under blue LED (470 nm) irradiation (Figure 1, and for the ^1^H-, ^13^C-NMR, and MS, please see Appendix A). A yield of 43.4% was achieved when the powder of CR was added as the catalyst (3% of the mass of *o*-phenylenediamine) (Table 1). In contrast, only a trace amount of product was formed when there was no light source, short irradiation time (2 h), or no catalyst, indicating that the powder of CR possessed photocatalytic activity. Isoquinoline alkaloids are the main active ingredients of CR. Thus, the total alkaloid was extracted (yield 8%) from CR powder (1 g). The catalytic efficiency of total alkaloids was 58.2% (Table 1), indicating that the alkaloid is the major active ingredient responsible for the catalyzing capacity of CR.

### 2.2. HPLC Analysis of the Total Alkaloid and Prescreening of Coformers

In order to characterize the catalytically active components in the total alkaloid, we performed HPLC analysis. The chromatogram of the total alkaloid showed seven major peaks (Figure 2a). The methanol solution of the total alkaloid was tested against a panel of coformers possessing an acid group or π-electron system, i.e., carboxylic acids (fumaric acid (i) [9] and succinic acid (ii) [10]), organic base (theophylline (iii) [11]), and sulfonic acids (1,5-naphthalenedisulfonic acid (NDS, iv) and 2-naphthalenesulfonic acid (v)). After 0.5 h, only tube (iv) that had NDS added showed precipitate (Figure 3). After 2.0 h, more precipitate was formed in tube (iv), while only a small amount of precipitate appeared in tube (i). However, other tubes still remained clear after 2.0 h (Figure 3). The resulting precipitate in tube (iv) was centrifuged and dissolved in methanol. After filtration through a 0.22 μm membrane, it was analyzed by HPLC in the same condition as the total alkaloid (Figure 2b). We found that the precipitate formed by NDS contained all the seven peaks as the original total alkaloids, except for the relatively increased intensity of peaks **2**–**7** (Figure 2c), indicating that NDS has strong interactions with all these peaks. Although the relative intensity for peak **1** remained largely unchanged as compared to peaks **2**–**7**, this component still exhibited sufficient interaction with NDS (Figure 2c).

Then, all the seven peaks in the total alkaloid chromatogram generated by semi-preparative HPLC were collected individually and concentrated in a SpeedVac to afford compounds **1**–**7**. A certain amount of NDS (based on the stoichiometric ratio from ^1^H-NMR analysis of the precipitate) in methanol solution was added to the residue of each compound. The mixed solutions were kept at room temperature to grow single crystals. It is noteworthy that coformers can be efficiently pre-screened for their binding affinity to the analysis by a comparison of the high performance liquid chromatography (HPLC) chromatograms of the crude extract before and after the addition of coformers.

### 2.3. Identification of the Individual Component by Co-Crystallization

NDS was found to be a suitable coformer in our pre-screening test. The precipitate formed by adding NDS to the individual peak solution (collected through semi-preparative HPLC. Peaks **3** and **4** are two close peaks. Collection of these two peaks should be undertaken very carefully, and only narrow peak time could be collected.) was filtered and dissolved in methanol, and new co-crystals of NDS with the seven peaks were obtained at room temperature in stoichiometric quantities through a single-step crystallization.

^1^H-NMR spectrum of the precipitate formed by adding NDS to compound **1** showed 1:2 stoichiometric ratio. The up-field shifted resonance of Ha (−0.05 ppm), Hb (−0.07 ppm) and Hc (−0.01 ppm) on the benzene ring of NDS (Table 2) indicated the formation of inclusion complex (Figure 4a), which was also supported by NOESY spectrum, i.e., NOE correlations were observed between NDS and **1** (Appendix A). Furthermore, inclusion complex of (NDS) ⊃**1**_2_ (the ⊃ symbol denotes inclusion) formed well-defined single crystals by slow evaporation of the methanol solution. Single-crystal X-ray diffraction data revealed that (NDS) ⊃**1**_2_ crystallized in the triclinic *P1* space group (Table 3). Compound **1** (C_20_H_24_NO_4_^+^) is a quaternary benzylisoquinoline alkaloid of the aporphine structural subgroup with a tetracyclic ring system bearing two hydroxy groups at the C-1 and C-11, two methoxy groups at the C-2 and C-10 positions, and two methyl groups on the nitrogen atom. From the Flack parameter calculated by the Parsons’ method [*x* = 0.06(3)], the absolute configuration of compound **1** was also determined (Figure 4b,c). By a literature search, compound **1** was identified as the known natural product α-magnolflorine [12]; however, its crystal structure and absolute configuration were established here for the first time. In a previous study [12], the work of Chen and co-investigators confirmed the presence of two isoforms of magnolflorine that exhibit different pharmacological activities. Our co-crystal X-ray analysis directly secured both the relative and absolute structure of compound **1**.

The asymmetric unit consist of one NDS, one methanol, two water and two molecules of **1** (Figure 4d), which showed the same structures with slightly different orientation of the methoxy group at C-10. Though there were three solvent molecules, none of them were disordered. The crystalline 1:2 ratio confirmed the result from NMR analysis. NDS and **1** was connected by various intermolecular C-H···O interactions (Appendix A) and C-H···π interactions (Appendix A) into a three-dimensional network (Figure 4e).

Similarly, the ^1^H-NMR spectrum of the precipitate formed by adding NDS to compounds **2** or **3** (Figure 5(2a,3a)) showed a 1:2 stoichiometric ratio. The Ha, Hb, and Hc of NDS demonstrated significant up-field shifts (−0.06, −0.08, and −0.02 ppm, respectively, when added to **2** or **3** (Table 2)) (Appendix A), indicating strong intermolecular interactions. Formation of inclusion complexes was also supported by 2D NOESY spectrum, i.e., NOE correlations were observed between NDS and **2** or **3** (Appendix A). Compounds **2** and **3** formed well-defined co-crystals with NDS. Single crystal structure analysis revealed that (NDS) ⊃ **2**_2_ and (NDS)_1/2_ ⊃ **3** crystallized in the monoclinic space groups *Cc* and P21/n, respectively (Table 3). Figure 5(2b) and Figure 6a showing the X-ray structure and packing diagram, respectively, for **2** and (NDS) ⊃ **2**_2_; and Figure 5(3b) and Figure 6b showing the X-ray structure and packing diagram, respectively, for **3** and (NDS)_1/2_ ⊃ **3**). Compound **2** bears a quaternary isoquinoline alkaloid skeleton with a 9-hydroxy-1, 10, 17-trimethoxy substitution pattern (C_20_H_20_NO_4_^+^). Compound **3** is similar to **2** (C_20_H_20_NO_4_^+^), with a different substitution pattern (10-hydroxy-1,9,17-trimethoxy). The asymmetric unit of (NDS) ⊃ **2**_2_ is composed of an NDS molecule, two compound **2** molecules, and two methanol molecules (without disorder) (Figure 5(2c)). In contrast, the asymmetric unit of (NDS)_1/2_ ⊃ **3** is composed of half an NDS molecule (located at the inversion center) and one molecule of **3** (same stoichiometric ratio 1:2 as **2**) but no solvent molecule (Figure 5(3c)). By a literature search, compound **2** was identified as the known natural product columbamine, which was reported to suppress hepatocellular carcinoma cells through down regulation of PI3K/AKT, p38, and ERK1/2 MAPK signaling pathways [13], while compound **3** was identified as the known natural product jatrorrhizine, which was reported to be a potent antagonist against the transmembrane protease to impact COVID-19 drug therapy [14].

Compounds **4** and **5** (Figure 5(4a,5a)) were two minor peaks. Especially, **4** is the shoulder of **3** in the chromatogram (Figure 2b). These two compounds were easily skipped in classic natural product chemistry. Precipitates occurred when NDS in methanol was added to **4** or **5**. ^1^H-NMR analysis on the precipitates revealed up-field chemical shifts for Ha, Hb, and Hc (−0.07, −0.08, and −0.02 ppm when NDS was added to **4**, and −0.06, −0.08, and −0.02 ppm when NDS was added to **5**) (Table 2), indicating the strong interactions between NDS and **4** or **5** (Appendix A), which were also confirmed by the NOESY spectra of these two inclusion complexes (Appendix A). Then, the precipitates were filtered, and the filtrates were subjected to crystal growth. Crystals appeared within two days. Single crystal X-ray analysis revealed that both (NDS)_1/2_ ⊃ **4** and (NDS) ⊃ **5**_2_ crystallized in the same triclinic space group *P-1* (Table 3). Figure 5(4b) and Figure 6c showing the X-ray structure and packing diagram, respectively, for **4** and (NDS)_1/2_ ⊃ **4**; and Figure 5(5b) and Figure 6d showing the X-ray structure and packing diagram, respectively, for **5** and (NDS) ⊃ **5**_2_). The asymmetric unit of (NDS)_1/2_ ⊃ **4** is composed of half an NDS molecule at the inversion center and one molecule of **4** (stoichiometric ratio 1:2) (Figure 5(4c)). In contrast, the asymmetric unit of (NDS) ⊃ **5**_2_ consists of an NDS molecule and two molecules of **5** (same stoichiometric ratio 1:2 as **4**) (Figure 5(5c)). Compound **4** (C_20_H_18_NO_4_) bears a pentacyclic protoberberine skeleton with two methoxy groups at C-9 and C-10, and a methylenedioxy group at C-1 and C-17. Accordingly, compound **4** was identified as the known natural product epiberberine, which was reported to be a potent inhibitor against the ureases from Helicobacter pylori [15]. The skeleton of **5** is the same as that of **4**, but the two methoxy groups at C-9 and C-10 in **4** were replaced by a methylenedioxy group in **5**. Accordingly, compound **5** (C_19_H_14_NO_4_) was identified as the known natural product coptisine, which was found to inhibit plasmodium falciparum dihydroorotate dehydrogenase, a promising drug target for antimalarial chemotherapy [16]. This compound is also a promising candidate agent for prevention of diabetic neuropathy [17].

Compounds **6** and **7** (Figure 5(6a,7a)) were the last two peaks in the precipitate chromatogram (Figure 2b). The ^1^H-NMR spectrum on the precipitate formed by adding NDS to **6** or **7** showed up-field chemical shifts for Ha, Hb, and Hc (−0.07, −0.10, and −0.03 ppm for **6**, and −0.06, −0.09, and −0.02 ppm for **7** (Table 2)) (Appendix A), indicating significant interactions, which were again confirmed by 2D-NOESY spectra of these two inclusion complexes (Appendix A). Both inclusion complexes of (NDS)_1/2_ ⊃ **6** and (NDS)_1/2_ ⊃ **7** formed well-defined single crystals by slow evaporation of the methanol solution at room temperature. Single-crystal X-ray diffraction data revealed that (NDS)_1/2_ ⊃ **6** and (NDS)_1/2_ ⊃ **7** crystallized in the same triclinic P-1 space group (Table 3). Figure 5(6b) and Figure 6e showing the X-ray structure and packing diagram, respectively, for **6** and (NDS)_1/2_ ⊃ **6**; and Figure 5(7b) and Figure 6f showing the X-ray structure and packing diagram, respectively, for **7** and (NDS)_1/2_ ⊃ **7**). Compound **6** (C_21_H_22_NO_4_^+^) is a quaternary protoberberine alkaloid with four methoxy groups at C-1, C-9, C-10, and C-17 positions. Compound **7** (C_21_H_22_NO_4_^+^) has the same skeleton as **6**, but the methoxy groups at C-9 and C-10 were changed to a methylenedioxy group. By a literature search, compounds **6** and **7** were identified as the known natural product palmatine and berberine (C_20_H_18_NO_4_^+^), respectively. Both compounds were active on the α2-adrenergic receptor with IC_50_ at the nanomole levels, and inhibit the multiplication of bacteria, fungi, and viruses [18]. Both the asymmetric units of (NDS)_1/2_ ⊃ **6** and (NDS)_1/2_ ⊃ **7** contain half an NDS molecule located at the inversion center and one molecule of compound **6** or **7** (Figure 5(6c,7c)). Due to the strong C-H···O and C-H···π interactions, no disorder was observed in these crystal structures (Appendix A).

### 2.4. NDS Decreases the Required Amounts of Alkaloids for Crystallization to Nanoscale

NDS greatly decreases the required crystallization amounts of alkaloids. Compound **3** (jatrorrhizine) was used as an example. When **3** (0.5 μg/μL, 0.7 μL) was layered with NDS (0.5 μg/μL, 0.7 μL) in a capillary tube (diameter 0.3 mm), co-crystals formed within 1 h. In contrast, no crystal was observed when keeping **3** alone. Thus, the amount required for the crystallization of **3** was only 0.5 μg/μL × 0.7 μL = 350 ng (Appendix A). It is noteworthy that compound **4** is the shoulder of **3** in the chromatogram (Figure 2b). Though we could only purify small amounts of **4** and **3** (baseline separation of **3** and **4** was not achieved), with the help of NDS, only a nanoscale of compounds was needed to be obtained for pure enough co-crystals for the SCD analysis.

### 2.5. Photocatalysis by the Individual Alkaloids

Through offline HPLC-CC-SCD analysis, compounds **1**–**7** in the active total alkaloid were identified as magnoflorine (Figure 4b), columbamine (Figure 5(2a)), jatrorrhizine (Figure 5(3a)), epiberberine (Figure 5(4a)), coptisine (Figure 5(5a)), palmatine (Figure 5(6a)), and berberine (Figure 5(7a)), respectively. To explore the active component responsible for the catalytic efficiency of the herbal powder and total alkaloid, we compared the yield of individual alkaloids of CR in the photocatalytic reaction. Surprisingly, all alkaloids had catalytic ability with a yield in the range 54.6–82.1%. Compound **1** (magnoflorine), bearing a short conjugation system, showed the lowest catalytic efficiency (54.6%), which is higher than the powder of CR but slightly lower than total alkaloid (58.2%). Compounds **2**–**7**, possessing the isoquinoline alkaloid skeleton with a long conjugation system, showed excellent catalytic efficiency (71.2–82.1%) (Table 1).

## 3. Discussion

Traditionally, the synthesis of benzothiazoles involved the condensation of phenylenediamine and its derivatives with carboxylic acids or aldehydes [19]. In recent years, catalyzed synthesis using different Lewis acids as the catalysis, e.g., SnCl_2_, appeared to improve the yield [20]. Visible light as a convenient and renewable clean energy source has attracted increasing interest in organic photosynthesis [21]. Compared with traditional synthesis and heavy metal catalysis, visible light catalysis meets the requirements of green chemistry and environmental friendliness. Very recently, synthesis of benzothiazoles using fluorescein as an efficient photocatalyst under visible light was reported in high to excellent yields [22]. However, photocatalysts using herbal materials were less reported.

CR is derived from the roots of Coptis chinensis, a species of flowering plant, which is traditionally used as an antibacterial herb [23]. This herb is a rich source of isoquinoline alkaloids with conjugated chromophores, e.g., berberine and palmatine. Besides the bactericidal activity, it is noteworthy that this class of alkaloids were found to show strong photodynamic effects. Berberine is an efficient photosensitizer. Its association with photodynamic therapy may be a potential anticancer treatment strategy for cervical cancer [24]. The DNA damage generated by a combination of berberine with UVA irradiation induced a significant blockage of EAC cells (Ehrlich ascites carcinoma) in the S and G2/M phases and the decrease of cell proliferation after 24 h of treatment [25]. Photodynamic therapy of palmatine exhibited a potent phototoxic effect in cell proliferation and produced cell apoptosis [26].

Considering the strong photodynamic effect of isoquinoline alkaloids and the conjugated skeleton, in this context, we demonstrated that the raw material Coptidis rhizome and the major isoquinoline alkaloids could be used as visible light catalysis for the green synthesis of benzimidazole. Furthermore, a HPLC based pre-screening approach was established for the selection of a suitable coformer. The subsequent single crystal X-ray diffraction analysis enables the rapid structural analysis of the photoactive molecules in the context of a hydrogen-bonded network without the necessity for massive phytochemical isolation.

Notably, these alkaloids possess a relatively low concentration level in plants, which makes their exact structure hard to analyze via spectral analysis in a short time. The most accurate method to determine the three-dimensional structure of molecules is single-crystal X-ray crystallography. However, this method requires the molecules to be crystallized and single crystals should be obtained before analysis. Several approaches are being pursued to address this limitation: (i) Host–guest chemistry. Cucurbituril [27], porphyrins [28], crown ether [29], calixarenes [30], cyclodextrins [31], co-crystals [32,33], MOF [34], and the recently reported mechanically interlocked structures [35] have proven abilities to capture a broad range of different organic molecules. (ii) Hydrogen-bonded frameworks. Guanidinium organosulfonates were found to capture various guest molecules of interest through the formation of hydrogen-bonded frameworks [36]. (iii) Crystal sponge. This method uses a metal–organic framework to absorb guest compounds, which were efficiently trapped and concentrated in the periodic array of the binding sites of the porous complexes either through intermolecular interactions [37] or coordinative alignments [38]. Furthermore, recently, scientists found that an ultrastable π–π stacked porous organic molecular framework can also be used as a crystalline sponge [39]. (iv) DNA duplex sequence. This method used an oligonucleotide sequence to capture the guest molecules as minor groove binder [40].

In this context, we successfully discovered a coformer to help arrange these low concentration molecules into a highly ordered state in the co-crystals. Through X-ray crystallography, these alkaloid guests’ detailed structures were clearly and rapidly elucidated, and the specific interactions between the co-crystal were proved. 1,5-Naphthalenedisulfonic acid (NDS) as a new family member of coformer can be regarded as an ideal candidate for the structural determination of certain alkaloids. It is noteworthy that the calculated electrostatic potential surface of NDS showed a strong negative charge area around the 1,5-disulfonate groups, which were consistent with the strong interactions with the alkaloids in CR with positive nitrogen atoms (Figure 7).

## 4. Materials and Methods

### 4.1. Extraction of Total Alkaloids from CR

Commercially purchased CR herb was crushed and sieved to obtain CR powder. The powder (1 g) was extracted by ultrasonic treatment with 95% EtOH at room temperature three times. The solution was filtered and concentrated under reduced pressure to obtain the crude extract. Then, an appropriate amount of 2% HCl solution was added to solve the crude extract and adjust the pH = 4. The solution was filtered, and the filtrate was alkalized with NaOH to pH = 9, which was then extracted with dichloromethane. Finally, the solvent was removed under reduced pressure to afford the total alkaloids of CR (80 mg).

### 4.2. Separation of Individual Alkaloids

The total alkaloids were subjected to semi-preparative HPLC on a Wufeng LC-100 system (Shanghai, China) with column: ultimate XB-C18 column (250 × 10.0 mm, 5 μm); mobile phase: 0.1% formic acid water (A) acetonitrile (B); gradient: 5 → 15% B in 0–10 min; 15 → 25% B in 10–15 min; 25% B in 15–35 min; flow rate: 3.0 mL/min, detection: UV at 278 nm. The collected solution of each peak was concentrated in a SpeedVac to afford compounds **1**–**7**.

### 4.3. Co-Crystallization

NDS was added to the solution of individual peaks of the total alkaloids. The precipitate formed by adding NDS was filtered and dissolved in methanol, and co-crystals of NDS with the seven peaks were obtained at room temperature.

### 4.4. Single-Crystal X-ray Diffraction Analysis

All single-crystal measurements were performed on a Rigaku Oxford diffractometer (Rigaku, Co. Tokyo, Japan) using CuKα (λ = 1.54056 Å) radiation. The structures were solved by direct methods (SHELXTL-2014) and refined by full-matrix least-squares on *F*^2^. Crystallographic data in standard CIF format were deposited with the Cambridge Crystallographic Data Centre. The CCDC numbers are shown in Table 3. Copies of the data can be obtained, free of charge, on application to CCDC, 12 Union Road, Cambridge CB2 1EZ, U.K. (fax: (+44)1223-336033; e-mail: deposit@ccdc.cam.ac.uk).

### 4.5. Electrostatic Potential Surfaces

All structures (generated by Chemdraw) were fully optimized at the B3LYP/6-31G(d) theoretical level. Vibrational analyses at the same level of theory were performed to confirm each stationary point to a local minimum. Electrostatic potential surface (ESP) values were calculated at the B3LYP/6-31G(d) level in the Gaussian 09 program. The electrostatic potential surfaces (ESP) of computed species were generated with Gaussview 6.0.

### 4.6. Synthesis of Benzimidazole Using CR and the Alkaloids as Catalysts

Benzaldehyde (1a, 0.1 mmol), *o*-phenylenediamine (1b, 0.1 mmol), acetonitrile (1.5 mL), and catalysts (including the herbal power, total alkaloid, and individual alkaloid, 3%) were added in a tube (10 mL). The open-air reaction tube was placed under a 5 W blue LED lamp and stirred at room temperature for 12 h. The reaction was extracted with ethyl acetate, and the organic layer was washed with water. The organic layer was saturated with sodium chloride and separated. Then the organic layer was dried with anhydrous Na_2_SO_4_ and concentrated under reduced pressure. The crude product was purified through silica gel column chromatography (petroleum ether: ethyl acetate = 2:1) to afford a light-yellow product (1c). The yield was calculated by the HPLC peak area using the purified 1c as the standard. ^1^H NMR (400 MHz, DMSO) δ 12.93 (s, 1H), 8.23–8.15 (m, 2H), 7.65–7.44 (m, 5H), 7.25–7.15 (m, 2H). ^13^C NMR (100 MHz, DMSO) δ 151.2, 130.1, 129.8, 128.9, 126.4, 122.1, 115.1.

## 5. Conclusions

In summary, our pre-screening with HPLC analysis successfully demonstrated that NDS can act as an excellent coformer for the rapid co-crystallization with the major alkaloids in CR. The structures of the seven new co-crystals were then determined by X-ray crystallographic analysis. This work provides a new efficient workflow featuring the offline HPLC/CC/SCD strategy and no massive phytochemical work was conducted. NDS greatly decreases the required crystallization amounts of alkaloids to the nanoscale. Furthermore, all the identified alkaloids were firstly found to show photocatalytic ability with the yields in the range 54.6–82.1%. The photocatalytic ability was found to be related to the conjugation system. Compound **1** (magnoflorine), bearing a short conjugation system, showed the lowest catalytic efficiency, while Compounds **2–7**, possessing a long conjugation system, showed excellent catalytic efficiency. Accordingly, besides the new HPLC/CC/SCD workflow, this work sheds light on the photocatalytic potential of natural products, though the biological effects of natural products were widely reported.

## Data Availability

The data presented in this study are available in the Appendix A.

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
