# Peer review of "Identification of Photocatalytic Alkaloids from Coptidis Rhizome by an Offline HPLC/CC/SCD Approach"

_molecules, 2022, doi:10.3390/molecules27196179_

Round 1
Reviewer 1 Report
The authors applied HPLC to separate components in Coptidis rhizome and mix the collected fractions with NDS allowing co-crystallization followed by the crystal analysis with SCD. This leads to the achievement of the seven alkaloid structures. The structure of compound 1 in this sample has been reported for the first time (Lines 159-160) albeit with the lowest catalytic efficiency compared with the others. Significant supporting data were provided. My concerns are listed below.
- HPLC peaks of several compounds coeluted (especially compounds 3 and 4) as shown in Figure 2. How the authors obtained pure enough crystals for the SCD analysis?
-The abbreviation of HPLC-CC-SCD is misleading since this in analytical areas generally means hyphenation (with the online/automation processes). After reading the experimental part, it appears that the process was performed manually from the HPLC fraction collection to the CC and SCD. It is more suitable to use other abbreviation such as “HPLC/CC/SCD” or “HPLC, CC and SCD” separately. Please correct this throughout the manuscript. At least, the suggested abbreviations or the word “offline” should be added into the title and please define HPLC-CC-SCD as (high performance liquid chromatography followed by co-crystallization and single crystal diffraction).
-(a) (b) and (c) labels in Figure 2 are missing.
-Parenthesis system as well as English in Lines 98-100 is confusing. Please correct/simplify this.
-Lines 100-101: … After 0.5 h, only the tube (d) that was added NDS showed precipitate (Figure 3b). What is the meaning of 3b in Figure 3? (a) (b) and (c) labels in this figure are also missing. This labelling is also confusing with the a-e indicating each co-crystal reagent. This can be replaced with the other labelling system such as 1-5 or i-v.
-Line 118: HPLC of the precipitate obtained after adding NDS and Line 103: The resulting precipitate in tube (d) was analyzed by HPLC … The detail should be provided how the authors analyzed “precipitate” in HPLC, e.g. dissolving in which solvent, with or without filtering?
-The interactions between a co-former and the analytes were investigated according to pre-screening HPLC test which is considerably effective (e.g. Lines 130-132, Lines 383-385). However, how could the authors prove the specific interactions between the co-crystal or any interaction could result in effective co-crystal formation? Could it be just that the target compounds had poor solubility in the medium used to wash the co-crystal prior to being dissolved for the HPLC analysis?
-The precipitate formed by adding NDS to the individual peak solution Lines 134-135. How could the peaks be individual with the coelution (e.g. peaks 3 and 4 in Figure 2)?
-The interactions such as C-H···O interactions, C-H···p should be clearly identified in each figure in Supporting Information since there are more than one possible interaction positions.
- a,b,c with the arrows in each figure in Figure 6 should be defined.
-Unit of the electrostatic potential in Figure7 should be provided.
-Please keep consistent in capitalization in Table 3, e.g. triclinic or Triclinic?
-Have the molecular geometries been optimized before the calculation of the electrostatic potential surfaces using Gaussian? Or obtained from the single crystal? this should be informed in section 4.5.
Author Response
Sept. 15, 2022
Professor Felicia Yao
Associate Editor
Molecules
Re: Manuscript-1929855
Dear Professor Felicia Yao
Thank you very much for your letter dated Sept. 12, 2022. We have revised the manuscript based on the reviewer’s comments.
The comments have been addressed as follows.
Reviewer 1
Comments to the Author
The authors applied HPLC to separate components in Coptidis rhizome and mix the collected fractions with NDS allowing co-crystallization followed by the crystal analysis with SCD. This leads to the achievement of the seven alkaloid structures. The structure of compound 1 in this sample has been reported for the first time (Lines 159-160) albeit with the lowest catalytic efficiency compared with the others. Significant supporting data were provided. My concerns are listed below.
- HPLC peaks of several compounds coeluted (especially compounds 3 and 4) as shown in Figure 2. How the authors obtained pure enough crystals for the SCD analysis?
Response:
All the seven peaks in the total alkaloid chromatogram generated by semi-preparative HPLC were collected individually and concentrated in SpeedVac to afford compounds 3 and 4. With the help of NDS, only nanoscale of compounds was needed to obtained pure enough cocrystals for the SCD analysis. In section 2.4, we added the following sentence so as to make this point clearer.
“It is noteworthy that compound 4 is the shoulder of 3 in the chromatogram (Figure 2b). Though we could only purify small amounts of 4 and 3 (baseline separation of 3 and 4 was not achieved), with the help of NDS, only nanoscale of compounds was needed to obtained pure enough cocrystals for the SCD analysis.”
- The abbreviation of HPLC-CC-SCD is misleading since this in analytical areas generally means hyphenation (with the online/automation processes). After reading the experimental part, it appears that the process was performed manually from the HPLC fraction collection to the CC and SCD. It is more suitable to use other abbreviation such as “HPLC/CC/SCD” or “HPLC, CC and SCD” separately. Please correct this throughout the manuscript. At least, the suggested abbreviations or the word “offline” should be added into the title and please define HPLC-CC-SCD as (high performance liquid chromatography followed by co-crystallization and single crystal diffraction).
Response:
Following the suggestions of the reviewer, we have changed the HPLC-CC-SCD to HPLC/CC/SCD, and change the title of this article from "Identification of photocatalytic alkaloids from Coptidis rhizome by HPLC-CC-SCD approach" to "Identification of photocatalytic alkaloids from Coptidis rhizome by offline HPLC/CC/SCD approach".
- (a) (b) and (c) labels in Figure 2 are missing.
Response:
We have changed the label in Figure 2 to “(a) HPLC of the total alkaloids; (b) HPLC of the precipitate obtained after adding NDS; (c) Overlay of the chromatograms before (blue) and after (orange) adding NDS.”
- Parenthesis system as well as English in Lines 98-100 is confusing. Please correct/simplify this.
Response:
We have changed the labels in parenthesis to i, ii, iii, iv and v, respectively.
Lines 98-100 was changed to “i.e., carboxylic acids [fumaric acid (i)[9] and succinic acid (ii)[10], organic base [theophylline (iii)[11], and sulfonic acids [1,5-naphthalenedisulfonic acid (NDS, iv) and 2-naphthalenesulfonic acid (v)].”
- Lines 100-101: … After 0.5 h, only the tube (d) that was added NDS showed precipitate (Figure 3b). What is the meaning of 3b in Figure 3? (a) (b) and (c) labels in this figure are also missing. This labelling is also confusing with the a-e indicating each co-crystal reagent. This can be replaced with the other labelling system such as 1-5 or i-v.
Response:
We changed Figure 3b to Figure 3.
We followed the reviewer’s suggestion. Labels (a), (b), (c), (d) and (e) in Figure 3 were changed to i, ii, iii, iv and v, respectively.
- Line 118: HPLC of the precipitate obtained after adding NDS and Line 103: The resulting precipitate in tube (d) was analyzed by HPLC … The detail should be provided how the authors analyzed “precipitate” in HPLC, e.g.dissolving in which solvent, with or without filtering?
Response:
Yes, we added more details in the sentence.
“The resulting precipitate in tube (iv) was centrifuged and dissolved in methanol. After filtration through a 0.22mm membrane, it was analyzed by HPLC at the same condition as the total alkaloid (Figure 2b).”
- The interactions between a co-former and the analytes were investigated according to pre-screening HPLC test which is considerably effective (e.g.Lines 130-132, Lines 383-385). However, how could the authors prove the specific interactions between the co-crystal or any interaction could result in effective co-crystal formation? Could it be just that the target compounds had poor solubility in the medium used to wash the co-crystal prior to being dissolved for the HPLC analysis?
Response:
Thanks for the reviewer’s comments. Proton transfer, charge transfer, hydrogen bonding and p-p interactions can all be used for the design of co-crystals. All the specific interactions between the co-crystal can be proved by single crystal diffraction analysis.
In this manuscript, NDS bearing two sulfonic acid groups was used as the cofomer, and it was found to decrease the required amounts of alkaloids for crystallization to nanoscale.
We added more descriptions in lines 97, 261 and 336 to make these comments clearer.
- The precipitate formed by adding NDS to the individual peak solution Lines134-135. How could the peaks be individual with the coelution (e.g.peaks 3 and 4 in Figure 2)?
Response:
We changed “individual peak solution” to “individual peak solution (collected through semi-preparative HPLC. Peaks 3 and 4 are two close peaks. Collection of these two peaks should be very carefully, and only narrow peak time could be collected.)”
- The interactions such as C-H···O interactions, C-H···p should be clearly identified in each figure in Supporting Information since there are more than one possible interaction positions.
Response:
We have clearly showed the atom labels of the C-H···O and C-H···π interactions in each figure in Supporting Information.
- a,b,c with the arrows in each figure in Figure 6 should be defined.
Response:
a, b and c with the arrows in each figure means the observation direction. We added this description in the Figure legend.
- Unit of the electrostatic potential in Figure 7 should be provided.
Response:
The unit (a.u.) for the electrostatic potential is added.
A.u. of Electric Potential (a.u.) is a unit in the category of Electric potential. It is also known as atomic units. This unit is commonly used in the a.u. unit system. A.u. of Electric Potential (a.u.) has a dimension of ML2T-3I-1 where M is mass, L is length, T is time, and I is electric current.
We also added a note “Note: a. u. means the electrostatic potential unit” in the caption of Figure 7.
- Please keep consistent in capitalization in Table 3, e.g.triclinic or Triclinic?
Response:
We followed the reviewer’s suggestion and unified the word in Table 3. All “Triclinic” were changed to “triclinic”.
- Have the molecular geometries been optimized before the calculation of the electrostatic potential surfaces using Gaussian? Or obtained from the single crystal? this should be informed in section 4.5.
Response:
We have shown in section 4.5 the detailed calculation method. All structures were generated by Chemdraw and were fully optimized at B3LYP/6-31G(d) theoretical level. Vibrational analyses at the same level of theory were performed to confirm each stationary point to a local minimum. Electrostatic potential surfaces (ESP) values were calculated at the B3LYP /6-31G(d) level in the Gaussian 09 program.”
We made these changes in section 4.5.
In carrying out this revision, we have followed the recommendations from editor and reviewers and made suitable changes in the manuscript which were shown in yellow in the highlighted version. We would like to thank you and the reviewer for the comments.
With best wishes,
Sincerely yours,

Reviewer 2 Report
The manuscript entitled “Identification of photocatalytic alkaloids from Coptidis rhizome by HPLC-CC-SCD approach” is well organized and structured by Authors.
What the authors propose is well described and clear in the various sections. Especially with regard to Materials and Methods, and the results are accompanied by diagrams, illustrations and figures that are absolutely clear and easy to use for those reading the paper.
That the results obtained are useful and of high quality is absolutely self-evident.
The decision to consider minor revisions necessary is actually such precisely because only the introductory section does little to contextualise the Authors' excellent work. Everything else should be published without any changes in my humble opinion.
I think the manuscript is of very good quality both in the way it is written and, in the way, the experimental work was organised.
I therefore only have a few suggestions to make to the authors, which I summarise in the following points:
- expand the 'Introduction' section a little so as to better contextualise the whole experimental phase described below
- to improve the quality of the figures and photographs as far as possible, especially as regards their definition and sharpness, I am referring in particular to figures 1 and 3
I do not think any further corrections and/or modifications are necessary.
Author Response
Sept. 15, 2022
Professor Felicia Yao
Associate Editor
Molecules
Re: Manuscript-1929855
Dear Professor Felicia Yao
Thank you very much for your letter dated Sept. 12, 2022. We have revised the manuscript based on the reviewer’s comments.
The comments have been addressed as follows.
Reviewer 2
Comments to the Author
The manuscript entitled “Identification of photocatalytic alkaloids from Coptidis rhizome by HPLC-CC-SCD approach” is well organized and structured by Authors.
What the authors propose is well described and clear in the various sections. Especially with regard to Materials and Methods, and the results are accompanied by diagrams, illustrations and figures that are absolutely clear and easy to use for those reading the paper.
That the results obtained are useful and of high quality is absolutely self-evident.
The decision to consider minor revisions necessary is actually such precisely because only the introductory section does little to contextualise the Authors' excellent work. Everything else should be published without any changes in my humble opinion.
I think the manuscript is of very good quality both in the way it is written and, in the way, the experimental work was organised. I therefore only have a few suggestions to make to the authors, which I summarise in the following points:
- expand the 'Introduction' section a little so as to better contextualise the whole experimental phase described below
Response:
We slightly expanded the "Introduction" section.” The following sentences were added in the 'Introduction' section.
“However, current natural product research still faced challenges, e.g. traditional research strategies and low efficient protocols” in lines 35-36.
“High cost and low efficiency constitute major bottlenecks in the field of natural product chemistry, and development of new protocol is necessary” in lines 46-48.
- to improve the quality of the figures and photographs as far as possible, especially as regards their definition and sharpness, I am referring in particular to figures 1 and 3.
Response:
We followed the reviewer’s suggestions, and improved the resolution of Figures 1 and 3 as shown in pages 2 and 4, respectively.
In carrying out this revision, we have followed the recommendations from editor and reviewers and made suitable changes in the manuscript which were shown in yellow in the highlighted version. We would like to thank you and the reviewer for the comments.
With best wishes,
Sincerely yours,

Reviewer 3 Report
The identification of 7 alkaloids from Coptidis rhizome by HPLC-CC-SCD approach is reported. The approach proves to be efficient, but above all very practical.
It is recommended to accept the manuscript after some minor corrections
1. Lines 13, 34, 73, 80: write in the third person. (review the entire document)
2. Line 32: Natural products are secondary metabolites produced by living organisms. This idea is not correct... secondary metabolites are substances obtained from...
3. Lines 35-36: What challenges does research in natural products face? examples
4. The answer to the silver question on lines 46-47 is yes. It is not necessary to ask these kinds of questions.
Author Response
Sept. 15, 2022
Professor Felicia Yao
Associate Editor
Molecules
Re: Manuscript-1929855
Dear Professor Felicia Yao
Thank you very much for your letter dated Sept. 12, 2022. We have revised the manuscript based on the reviewer’s comments.
The comments have been addressed as follows.
Reviewer 3
Comments to the Author
The identification of 7 alkaloids from Coptidis rhizome by HPLC-CC-SCD approach is reported. The approach proves to be efficient, but above all very practical.
It is recommended to accept the manuscript after some minor corrections
- Lines 13, 34, 73, 80: write in the third person. (review the entire document)
Response:
We followed the reviewer’s suggestion. Lines 13, 34, 73 and 80 were written in the third person in the revised manuscript.
- Line 32: Natural products are secondary metabolites produced by living organisms. This idea is not correct... secondary metabolites are substances obtained from...
Response:
We have revised the sentence to “Natural products are produced naturally by any organism including primary and secondary metabolites.”
- Lines 35-36: What challenges does research in natural products face? Examples
Response:
We added “e.g. traditional research strategies and low efficient protocols” in line 36.
- The answer to the silver question on lines 46-47 is yes. It is not necessary to ask these kinds of questions.
Response:
We followed the suggestion of the reviewer and deleted "Can we speed-up the natural product research and find applications of the natural products other than biological effects? "
In carrying out this revision, we have followed the recommendations from editor and reviewers and made suitable changes in the manuscript which were shown in yellow in the highlighted version. We would like to thank you and the reviewer for the comments.
With best wishes,
Sincerely yours,
